



# Study of the equatorial and low-latitude TEC response to plasma bubbles during the solar cycle 24-25 over the Brazilian region using a Disturbance Ionosphere indeX

Giorgio Arlan Silva Picanço[1], Clezio Marcos Denardini[1], Paulo Alexandre Bronzato Nogueira[2], Laysa
Cristina Araujo Resende[1,3], Carolina Sousa Carmo[1], Sony Su Chen[1], Paulo França Barbosa-Neto[1],
Esmeralda Romero-Hernandez[4]

[1]National Institute for Space Research (INPE), São José dos Campos, SP, Brazil.
[2] Federal Institute of Education, Science and Technology of São Paulo (IFSP), Jacareí, SP, Brazil.
[3]National Space Science Center, Chinese Academy of Science (NSSC/CAS), Beijing, China.
[4]Universidad Autónoma de Nuevo León, Facultad de Ciencias Físico-Matemáticas (UANL/FCFM), Monterrey, Mexico.

*Correspondence to*: Giorgio A. S. Picanço (giorgio.picanco@inpe.br; giorgiopicanco@gmail.com)

**Abstract.** This work uses the Disturbance Ionosphere indeX (DIX) to evaluate the ionospheric responses to Equatorial Plasma Bubbles (EPBs) events from 2013 to 2020 over the Brazilian equatorial and low latitudes. We have compared the DIX variations during EPBs to ionosonde and All-Sky Imager data, aiming to evaluate the physical characteristics of these events. Our results show that the DIX was able to detect EPB-related TEC disturbances in terms of their intensity and occurrence times. Thus, the EPB-related DIX responses agreed with the ionosphere behavior before, during, and after the studied cases. Finally, we found that the magnitude of those disturbances followed most of the trend of solar activity, meaning that the EPB-related TEC variations tend to be higher (lower) in high (low) solar activity.

## 1 Introduction

The analysis of the Total Electron Content (TEC) is a useful technique for studying the ionosphere responses to several space weather phenomena. Thus, many studies have been conducted aiming to investigate the TEC variations during ionospheric disturbances originated from both external (e.g. magnetic storms) and internal drivers (e.g. gravity waves) (Chu et al., 2005; Nogueira et al., 2011; Astafyeva et al., 2015; Figueiredo et al., 2018). In this regard, the TEC is a referential measure of the ionosphere plasma density as a function of free electrons. This parameter can be obtained from the analysis of Global Navigation Satellite System (GNSS) signals and is thoroughly defined as the number





of free electrons in the ionized plasma contained along an imaginary tube with a cross-section of $1m^2$, whose ends are delimited by satellite in orbit and ground receiver (Kersley et al., 2004). Therefore, since the TEC is a good measure of the ionospheric plasma density, many efforts have been made to develop ionospheric indices for quantifying the TEC variations associated with the space weather

phenomena (Gulyaeva and Stanislawska, 2008; Sanz et al., 2014; Wilken et al., 2018; Jakowski and Hoque, 2019; Denardini et al., 2020a).

During quiet periods, the ionospheric plasma density tends to follow the regular day-to-day variation in terms of production/loss of electron-ion pairs, which is controlled mainly by the ratio between solar radiation incidence and the neutral atmosphere density (Kelley, 2009). It is worth

mentioning that the term 'quiet' has been used to describe both geomagnetic and ionospheric environments within a space weather context (Picanço et al., 2020). In this context, many space weather phenomena can affect the TEC calculation since it changes the distribution/concentration of plasma in the ionosphere. Among these phenomena are the Equatorial Plasma Bubbles (EPBs), which are plasma irregularities that originate in the equatorial ionosphere under the Rayleigh–Taylor instability (RTI)

condition and can propagate to higher latitude ranges (Kelley, 2009; Takahashi et al., 2015). One of the main characteristics of the EPBs is the plasma depletion that occurs in the form of geomagnetic field-aligned irregularities. Thus, the plasma density variation due to EPBs also changes the local ionospheric electrical conductivity, which causes depletions on TEC values due to the ionospheric refraction undergone by the GNSS signals (Otsuka et al., 2002; Takahashi et al., 2016).

Therefore, several ionospheric indices have been developed aiming to measure the ionospheric TEC effects due to EPBs. Among them, the Rate Of TEC Index (ROTI) is a technique for quantifying the amplitude of GNSS phase fluctuations and is commonly used as a direct measure of the EPBs occurrence (Pi et al., 1997; Cherniak et al. 2018; Carrano et al., 2019; Liu et al., 2019a, 2019b; Borries et al., 2020; Carmo et al., 2021). Another widely used index for monitoring the EPB effects is the

Scintillation Index (S4), which represents a value of ionospheric scintillation on GNSS signals mainly in the equatorial and low-latitude regions, and is capable of estimating the magnitude of the EPB-related scintillations (Aarons, 1982; Kintner et al., 2007). Furthermore, Denardini et al. (2020a) presented a





new version of the Disturbance Ionosphere Index (DIX), which provides a dimensionless value of the ionosphere perturbation degree during phenomena of both internal and external origin.

The DIX was firstly proposed by Jakowski et al. (2006) as a proxy for the TEC response due to magnetic storms. Later, new versions of this index were introduced aiming to analyze the phenomena originating from internal and/or external drivers (Jakowski et al., 2012; Wilken et al., 2018; Denardini et al., 2020a). Among them, the DIX version presented by Denardini et al., 2020a has been used for evaluating ionospheric disturbances over the Brazilian region. Additionally, Picanço et al. (2020)

presented a statistical evaluation of the DIX equation that Denardini et al. (2020a) proposed. The authors stated that the new DIX approach proved to be effective in detecting ionospheric disturbances due to internal sources and that the new method for calculating the DIX non-perturbed reference also made the index more sensitive to short-term TEC time variations. Afterward, Picanço et al. (2021) used the same DIX version to provide an analysis of the equatorial and low-latitude ionospheric response to

disturbed electric fields during extreme geomagnetic storms over the Brazilian region. Moreover, the authors discussed the DIX response to pre-storm effects and other phenomena, such as sporadic E layers. Denardini et al. (2020b) analyzed DIX maps calculated for South America during extreme geomagnetic storms and plasma bubbles. Besides the magnetic storm analysis, the authors associated local disturbances in DIX with the Equatorial Spread-F (ESF) observed over the same area of study.

However, as it was a preliminary study, the authors did not delve into the physical explanation of the DIX responses due to plasma bubbles. In addition, they presented only one EPB case that occurred along with the St. Patrick's Day magnetic storm period.

Thus, this study aims to evaluate the equatorial and low-latitude ionospheric responses to several plasma bubbles events distributed along the period from 2013 to 2020 over the Brazilian region.

Therefore, we used the DIX version presented in Denardini et al. (2020a) to characterize the ionospheric TEC disturbances due to the EPBs and compared it to the occurrence of ESF/plasma irregularities observed on ionosonde and All-Sky Imager OI 630 nm airglow data. Our results show that the DIX can detect EPB-related TEC disturbances in terms of their intensity and times of beginning and end. In short, the DIX responses were consistent with the ionosphere behavior before, during, and after the

occurrence of these phenomena. Finally, we found that the magnitude of TEC disturbances followed the



trend of solar activity, which means that the EPB-related TEC disturbances tend to be higher (lower) in high (low) solar activity.

## 2 Methodology

### 2.1 The Disturbance Ionosphere indeX (DIX)

We used the DIX methodology presented in Denardini et al. (2020a) to analyze the TEC variations during plasma bubbles events over the Brazilian equatorial and low-latitude ionospheric regions. In this regard, we calculated the DIX using TEC data obtained from the method presented in Seemala and Valladares (2011). The DIX for a specific GNSS station is defined by Equation (1):

$$DIX_k(t) = \left| \frac{\alpha(\Delta TEC(t)/TEC^{Qd}(t)) + \Delta TEC(t)}{\beta} \right|, \qquad (1)$$

where $\Delta TEC(t) = TEC(t) - TEC^{Qd}(t)$ is the difference between the current TEC value and the non-perturbed TEC reference for a given time $t$, $TEC_k$ is the current vertical TEC value, $TEC^{Qd}$ is the non-perturbed TEC reference obtained from a 3-hour centered moving average along the "reference day", $\alpha$ is the $TEC^{Qd}$ value at local midnight, being obtained for each period of analysis. Finally, $\beta$ is a latitudinal dependent term used to normalize the DIX value into a scale ranging from 0 to 5. Thus, it is relevant to mention that the "reference day" is defined as the geomagnetically quietest day of the period of study where no depletions greater than 20 TEC units over a 1-hour period were observed in the nighttime TEC. More details on the DIX calculation can be found in Denardini et al. (2020a) and Picanço et al. (2020).

The DIX is an index developed for representing the ionosphere perturbation degree during several types of phenomena (e.g. disturbed electric fields, plasma bubbles, sporadic E-layers). Therefore, we highlight that the ionospheric responses observed using this index are generally caused by the sum of concomitant events which can be caused by internal and/or external drives (Denardini et al., 2020a,b, Picanço et al., 2020, 2021). Thus, those studies demonstrate that the current DIX





methodology provides a straight response to space weather events, in which it is possible to estimate the intensity and duration of ionospheric disturbances.

The DIX provides a single and dimensionless value that represents the ionosphere degree of perturbation within a scale of ionospheric states. This scale ranges from 0 to 5 (Table 1), and represents the intensity of phenomena detected by DIX (Denardini et al., 2020a):


**Table 1.** The ionospheric states according to the DIX scale.

| DIX scale | Ionospheric state |
|---|---|
| $0 - 1$ | Quiet |
| $1 - 2$ | Weakly Disturbed |
| $2 - 4$ | Disturbed |
| $4 - 5$ | Exceptionally Disturbed |
| $> 5$ | Extremely Disturbed |

## 2.2 Instruments and data analysis

Figure 1 shows the geographic locations of the instruments used in this study. The red symbol
represents the site where the GNSS receiver, All-Sky Imager (ASI), and ionosonde (IONO) are available. Blue symbols represent sites with only GNSS receivers, while green symbols indicate ASI sites, and orange symbols represent IONO sites. The black lines identify the magnetic equator (0 degrees) and low latitudes (near ±15 degrees) for the year 2016 (midpoint of the period of study).

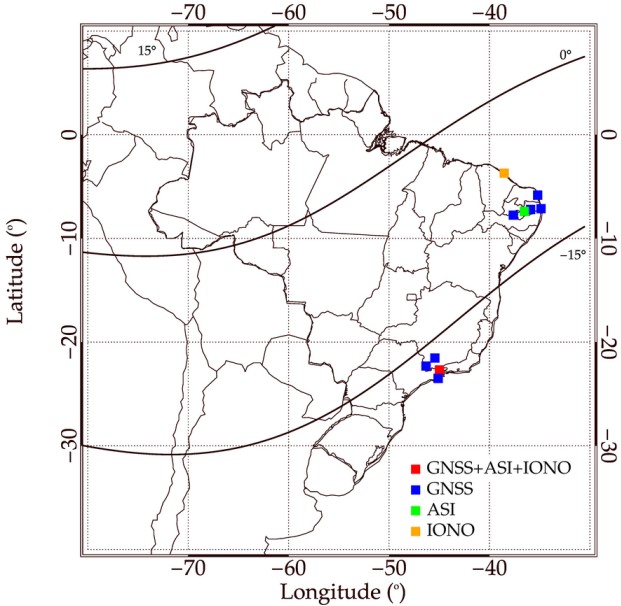

**Figure 1:** Map indicating the geographic locations of the instruments used in this study. The red symbol represents the site where GNSS receiver, All-Sky Imager (ASI), and ionosonde (IONO) are available. Blue symbols represent sites with only GNSS receivers, while green symbols indicate ASI sites, and orange symbols represent IONO sites. The black lines identify the magnetic equator (0 degrees) and low latitudes (near ±15 degrees) for the year 2016.

Table 2 shows more specific information about all the ten sites where the instruments used in this study are located. Therefore, we used data from eight GNSS receivers (Campina Grande, João Pessoa, Afogados da Ingazeira, Natal, Cachoeira Paulista, Ubatuba, Inconfidentes, Varginha, São João do Cariri e Fortaleza), two All-Sky Imagers (Cachoeira Paulista and São João do Cariri), and two ionosondes (Digisonde DPS-4D model: Cachoeira Paulista and Fortaleza). We emphasize that these

ionosondes operate with a 10/15 min sampling interval, in which the sounding frequencies range from 1.0 to 30.0 MHz (frequency step of 0.5 MHz) (Reinish et al., 2009).





**Table 2 -** Latitude, Longitude, and type of instrument present at the sites used in this study.

| Station (Code) | Type | Latitude (°) | Longitude (°) |
|---|---|---|---|
| Campina Grande (PBCG) | GNSS | -7.21 | -35.90 |
| João Pessoa (PBJP) | GNSS | -7.13 | -34.87 |
| Afogados da Ingazeira (PEAF) | GNSS | -7.76 | -37.63 |
| Natal (RNNA) | GNSS | -5.83 | -35.20 |
| Cachoeira Paulista (CHPI/CP/CAJ2M) | GNSS+ASI+IONO | -22.68 | -44.98 |
| Ubatuba (UBA1) | GNSS | -23.50 | -45.11 |
| Inconfidentes (MGIN) | GNSS | -22.31 | -46.32 |
| Varginha (MGVA) | GNSS | -21.54 | -45.43 |
| São João do Cariri (CA) | ASI | -7.39 | -36.53 |
| Fortaleza (FZA0M) | IONO | -3.71 | -38.54 |

Specifically, we divided the data sites into two regions: equatorial and low latitudes. Thus, we obtained data from four dual-frequency GNSS receivers (PBCG, PBJP, PEAF, and RNNA), one All-

Sky Imager (CA), and one ionosonde (FZA0M) for the equatorial region. The same was made for the low-latitudes region, where we collected data from four dual-frequency GNSS receivers (CHPI, MGIN, MGVA, and UBA1), one All-Sky Imager (CP), and one ionosonde (CAJ2M).

The GNSS data were used for obtaining the TEC values for the DIX calculation during selected plasma bubbles events from 2013 to 2020, which corresponds to the peak of solar cycle 24 up to the

ascending phase of the solar cycle 25. In this regard, we used the methodology presented in Seemala and Valladares (2011) for obtaining the TEC.

Moreover, we used the ASI data for identifying the occurrence of plasma bubbles over the two regions. This analysis was performed by observing plasma depletions on the OI 630 nm emissions images, which were also used to estimate the times of beginning and ending of the EPBs to support the

analysis of the DIX responses observed during these events.

Furthermore, we also compared the DIX and ASI data to the ionograms from the equatorial and low-latitude ionosondes, which were used to identify the presence of spread-F during the EPB events.



These ionograms were generated from the Standard Archiving Output (SAO) files obtained from the ionosondes. Thus, these comparisons were discussed in terms of the spread-F duration and intensity

during the EPB responses observed using DIX.

Finally, we compared the DIX responses to EPBs with the solar activity level as measured by the sunspot number during the period from 2013 to 2020 to evaluate the relation between the intensity of ionospheric disturbances with the season/solar activity.

## 3 Results and discussions

### 3.1 Validation of the DIX response to plasma bubbles

Figure 2 shows the time variations of the relative Slant TEC (STEC) (a), DIX (b), and ROTI (c) obtained on January 30, 2014, from the satellite PRN 3 observations of the following equatorial GNSS stations (left to right panels, respectively): PBCG, PBJP, PEAF, and RNNA. This figure also shows a

sequence of OI 630 nm airglow images taken at CA (d) during the same satellite PRN tracking time. Red arrows on these images indicate depletions associated with equatorial plasma bubbles.

We observe a DIX enhancement associated with the TEC fluctuations due to the plasma bubble occurrence. Specifically, the depletions from 01:00 to 03:00 UT that are poorly visible considering only the STEC curve (Figure 2a) appear as clear peaks in both DIX (Figure 2b) and ROTI (Figure 2c) during

the same time interval, in which both indices presented the highest values (DIX ~ 5 and ROTI ~ 0.6). This behavior is observed similarly but not identically over the four equatorial GNSS stations, which reinforces that the DIX provides a localized response to the TEC variations associated with the occurrence of ionospheric disturbances (as suggested by Picanço et al., 2021). Moreover, since the ROTI is one of the most reliable indices for detecting TEC changes due to ionospheric irregularities, the

simultaneous response of both indices is a good indication that the DIX is also capable of perceiving the presence of TEC fluctuations generated by plasma bubble events (Carmo et al., 2021). However, to ascertain the most-likely origin of these disturbances, we analyzed OI 630 nm airglow images taken at CA (Figure 2d), a site located at the center of the perimeter formed by the four equatorial GNSS

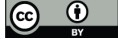

stations. These ASI images show the typical plasma bubble signature, a field-aligned depletion with
eastward motion (highlighted by the red arrows) (Sobral et al., 1985).

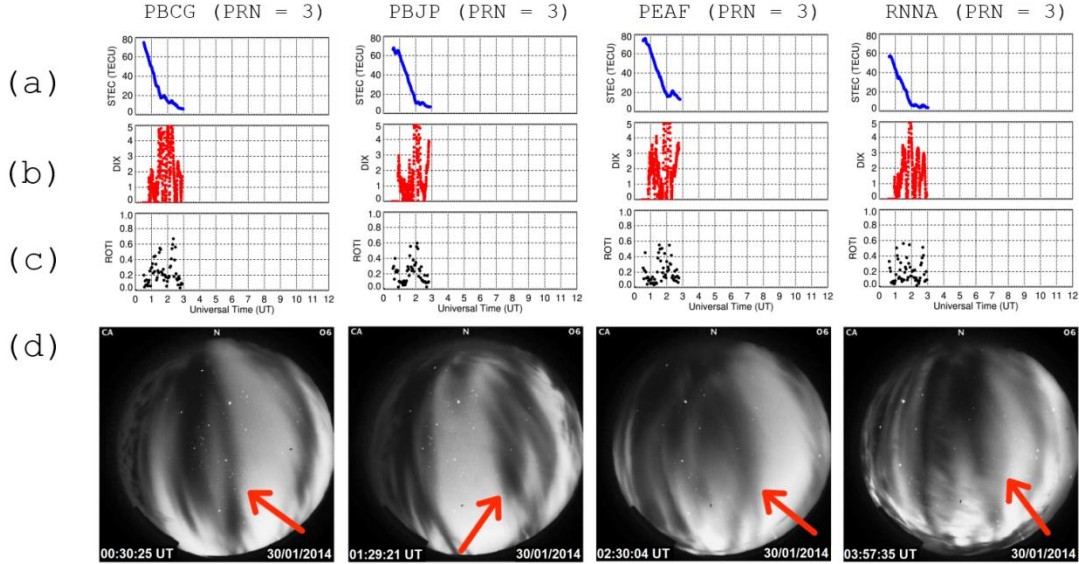

**Figure 2:** Relative STEC (a), DIX (b), and ROTI (c) for PBCG, PBJP, PEAF, and RNNA observations, along with
a sequence of OI 630 nm airglow images taken at CA (d) during the same satellite PRN 3 tracking time.

Figure 3 shows the time variations of the relative STEC (a), DIX (b), and ROTI (c) obtained for
30 January 2014 from the satellite PRN 23 observations of the following low-latitude GNSS stations
(left to right panels, respectively): CHPI, MGIN, MGV, and UBA1. Also in this figure, is presented a
set of OI 630 nm airglow images taken at CP (d) during the same satellite PRN tracking time. The red
arrows on these images indicate depletions associated with equatorial plasma bubbles.

From this figure, it is reasonable to state that there is a clear DIX response to the plasma bubble
occurrence, which is a result of the TEC depletions associated with this phenomenon. In this regard, we
observe some small-scale fluctuations within the STEC (Figure 3a) daily curve. However, analyzing
independently these variations appears not to be sufficient to evaluate the intensity of the TEC changes
due to the plasma bubbles event. Thus, by observing the projection of these disturbances in DIX (Figure
3b) and ROTI (Figure 3c), we noticed that both indices presented the highest values (DIX > 5 and ROTI





~ 1) in the period for 03:30 through 05:30 UT. Since DIX and ROTI raised to upper levels, it characterizes a considerably plasma bubble case.

In the next hours (05:30 to 09:00 UT), both indices start to follow a gradual decrease from 03:30 through 06:00 UT, reaching the zero level. Just like in the analysis of the equatorial stations, this behavior agrees with the expected characteristic of plasma bubbles, which are a typical phenomenon of the nighttime ionosphere (Pimenta et al., 2001). In addition, the same gradual behavior was observed in all four low-latitude GNSS stations, which presented DIX and ROTI enhancements at almost the same time. To verify the most likely origin of these disturbances, we observed OI 630 nm airglow images taken at CP (Figure 3d), a site located at the center of the perimeter formed by the four GNSS stations. These ASI images also showed the typical plasma bubbles signature (highlighted by the red arrows).

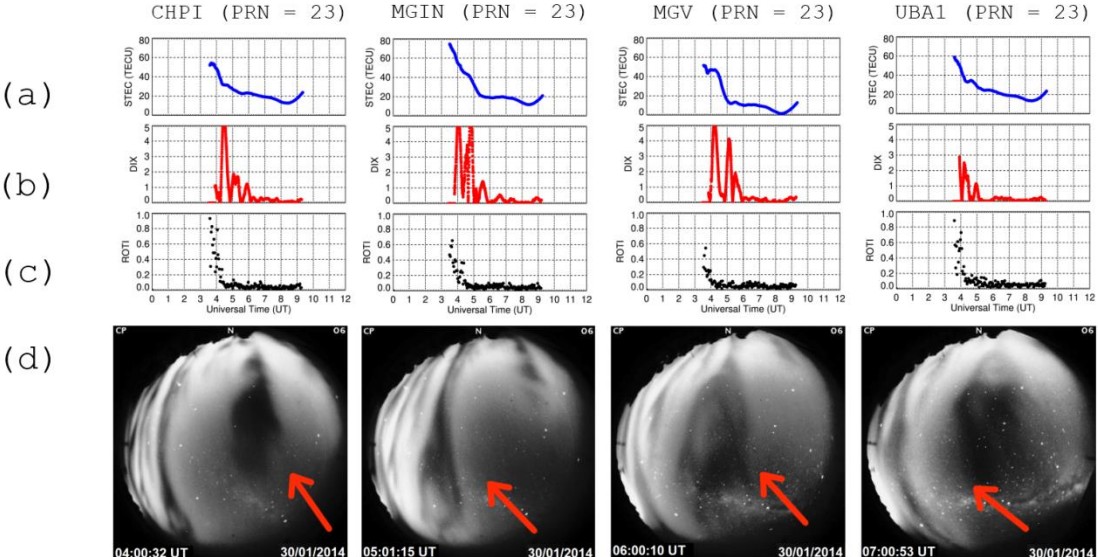

**Figure 3:** Relative STEC (a), DIX (b), and ROTI (c) for CHPI, MGIN, MGV, and UBA1 observations, along with a sequence of OI 630 nm airglow images taken at CP during the same satellite PRN 23 tracking time.

From those results, the DIX can be able to detect the ionosphere response to plasma bubbles in terms of the intensity of changes in its electronic density. In this context, case studies focused on





evaluating the ionospheric disturbances observed using DIX during plasma bubble events are presented

below.

### 3.2 Analysis of EPB-related ionospheric disturbances under high solar activity

In the present case study, we analyze the DIX responses during four selected plasma bubbles events

distributed along the period from around the peak (2013) to the midpoint of the descending phase

(2016) of solar cycle 24.

Figure 4 shows the time variations of the DIX (left panels) for the equatorial (PBCG, PBJP, PEAF,

and RNNA) and low-latitude (CHPI, MGIN, MGV, and UBA1) GNSS stations during the following

periods: 21-22 December 2013 (a), 1-2 February 2014 (b), 8-9 January 2015 (c), and 31 January–1

February 2016 (d). Along with the DIX graphs, this figure also includes OI 630 nm airglow images

(mid panels) indicating the occurrence of plasma bubbles over both CA and CP ASI sites. We also

present ionograms (right panels) from the FZA0M and CAJ2M ionosondes obtained at the same time

periods of DIX calculation. Orange bars on the DIX graphs indicate the time interval where bubble

signatures were seen on the available ASI data. Yellow bars indicate the interval where spread-F was

observed in the ionograms. Red arrows on the OI 630 nm airglow images indicate depletions associated

with equatorial plasma bubbles, while black arrows on the ionograms indicate the spread-F occurrence.

In the first case (Figure 4a), plasma bubble signatures were observed on the ASI and/or IONO

data at São João do Cariri from 22:30 to 05:30 UT on 21-22 December 2013. It is emphasized that the

same data were not available for Cachoeira Paulista at this period. Nonetheless, at around 23:00 UT, the

DIX over both latitude ranges started to increase in response to the TEC depletions. We notice that the

DIX over all stations responds to the EPB occurrence in the form of a gradual increase followed by a

decrease as the bubble/spread-F gets weak. The DIX over the equatorial stations shows two peaks, the

first one around 01:30 UT and the second around 03:20 UT. This behavior can be attributed to the

wave-form zonal propagation of these bubbles, which produces cyclic longitudinal TEC variations (as

reported in Takahashi et al., 2021). Following the same behavior, the DIX over low-latitude GNSS

stations shows fluctuations simultaneously, reaching the first peak at 02:10 UT and the second one at

~04:10 UT, which is 40 minutes later than in the equatorial stations. Therefore, we can state that DIX





was able to estimate the latitudinal time of propagation of the EPB-related plasma disturbances between the equatorial and low-latitude regions. Those disturbances remain within the weakly disturbed state (DIX under 2) in both latitude ranges and last for about five hours after their peaks. Notably, the low-latitude GNSS stations presented averagely higher DIX values than the equatorial ones during the plasma bubbles occurrence. Such effect can be explained by the presence of the Equatorial Ionization Anomaly (EIA) southern crest, which produces a large amount of plasma in this region, making the TEC percentage changes also higher (Abdu, 1997).

The second case (Figure 4b) shows the occurrence of plasma bubbles on 1-2 February 2014. This event was registered as spread-F on the FZA0M ionograms from 22:00 to 05:30. Ionosonde data from Cachoeira Paulista was not available during this period. However, plasma bubble signatures were seen on the available ASI data over this station from 23:30 to 05:30 UT. Nevertheless, in response to this plasma bubbles event, the DIX over equatorial GNSS stations started to rise at around 22:10 UT, followed by a peak at 01:20 UT (DIX under 3, disturbed state). Afterward, these disturbances lasted for about 4 hours after the peak. On the low-latitude stations, the DIX increases took place at almost the same time (first over the MGIN station). In contrast, the DIX peak occurred more than one hour later (~02:20 UT) and these disturbances lasted for about 5 hours after this peak (DIX under 3, disturbed state). Therefore, these disturbances started at almost the same time and followed the same gradual behavior as in the equatorial stations but lasted more. Since the DIX is an ionospheric index that responds to several type of ionospheric phenomena, we highlight that the disturbances observed over the equatorial stations were caused not only by the plasma bubbles event. These effects can be attributed to the slower ionospheric recombination occurring in the EIA region, which was confirmed by the TEC curve behavior (not shown here). Thus, it is possible that the ionosphere over low latitudes remained disturbed one hour after the end of the plasma bubbles event. However, since this analysis is beyond the scope of the present study, we intend to explain this in more detail in a further paper.

The third case (Figure 4c) shows plasma bubbles registered on the ASI and/or IONO data at São João do Cariri from 22:00 to 07:20 UT on 8-9 January 2015. Moreover, ionograms from the FZA0M ionosonde showed spread-F signatures from 00:00 to 01:30 UT and from 02:20 to 04:40 UT on 9 January 2015. It was not possible to see the bubble signatures on the OI 630 nm airglow images





obtained over CP since there was not enough sky visibility for this period. The DIX over all the stations started to rise before 21:00 UT. Nonetheless, these DIX increases precede the beginning phase of the plasma bubbles event, being related to local disturbances over both equatorial and low-latitude regions. These disturbances can be caused by changes in the E-region dynamo that occur during geomagnetic quiet periods, and cause the recombination of electrons and ions to be anticipated in comparison to the reference day used in the DIX calculation (Rishbeth, 1991; Fuller-Rowell et al., 1996; Kelley, 2009). Besides that, a peak at ~22:30 UT is observed on the DIX curve over all the equatorial stations, which occurs immediately after the EPB time of rising. In the low-latitude ionospheric region, the DIX peaks occur at 00:10 UT, within an interval of 1h40min after the peaks observed in the equatorial region. In these hours, we observed the spread-F in the ionograms from the CAJ2M ionosonde. After that, small DIX fluctuations are observed in both latitude ranges. However, these fluctuations tend to stop with the end of the plasma bubble event (after ~07:20 UT).

Lastly, the fourth case (Figure 4d) presents a plasma bubbles event registered on the ASI and/or IONO data from 21:50 to 07:40 UT at São João do Cariri, and from 23:30 to 04:20 UT at Cachoeira Paulista, between 31 January and 1 February 2015. The OI 630 nm airglow images from the CA ASI started to present bubble signatures at 21:50 UT on 31 January. After that, at 22:00 UT, the DIX over equatorial GNSS stations presented an enhancement, which evolved to a first peak at 23:59 UT. The same behavior is seen over low-latitude stations, but with these initial peaks happening 30 minutes later, at 00:29 UT. Then, two more peaks are seen on the DIX over the equatorial region; the first one at 01:50 UT and the second at ~04:30 UT. These peaks also occur in the low-latitudes DIX, at 02:40 UT and 04:50 UT, respectively. Additionally, these times represent a 50-min and 20-min delay concerning the equatorial region, which indicates that the average latitudinal EPB propagation time between the latitude ranges was about 33 minutes.





300



**Figure 4:** Time variations of the DIX (left panels) for equatorial and low-latitude stations, along with OI 630 nm
airglow images (mid panels) taken at CA and CP, and ionograms (right panels) showing the spread-F occurrence at FZA0M
and CAJ2M on 21-22 December 2013 (a), 1-2 February 2014 (b), 8-9 January 2015 (c), and 31 January–1 February 2016 (d).



### 3.3 Analysis of EPB-related ionospheric disturbances under low solar activity

In this case study, we analyze the DIX response to plasma bubbles during four selected events distributed throughout the period that starts around the cycle 24 descending phase (2017) up to the
ascending phase (2020) of solar cycle 25.

Figure 5 shows the time variations of the DIX (left panels) for the equatorial (PBCG, PBJP, PEAF, and RNNA) and low-latitude (CHPI, MGIN, MGV, and UBA1) GNSS stations during the following periods: 23-24 February 2017 (a), 5-6 January 2018 (b), 20-21 December 2019 (c), and 20-21 March 2020 (d). Additionally, this figure also shows OI 630 nm airglow images (mid panels) taken at the CA
and CP ASI sites, which indicate the occurrence of plasma bubbles around the same time periods of DIX calculation. Along with these figures, we present examples of ionograms (right panels) from the FZA0M and CAJ2M ionosondes in which we see the spread-F occurrence. Orange bars on the DIX graphs indicate the time interval where bubble signatures were seen on the available ASI data. Yellow bars indicate the interval where spread-F was observed in the ionograms. Red arrows on the OI 630 nm
airglow images indicate depletions associated with equatorial plasma bubbles, while black arrows on the ionograms indicate the spread-F occurrence.

In the first case (Figure 5a), plasma bubble signatures were observed on the ASI and/or IONO data from 22:00 to 07:20 UT at São João do Cariri on 23-24 February 2017, and from 00:30 to 04:50 UT at Cachoeira Paulista on 24 February 2017. In response to these events, the DIX over the equatorial
GNSS stations starts to rise at ~22:20 UT, following a gradual increasing behavior until it reaches to a peak at around 01:20 UT. However, we notice that this peak occurs approximately one hour earlier at the PEAF GNSS station, which is ~200 km west of PBCG. Since the EPBs move eastward in this case, we can state that DIX also responds to the zonal propagations of these phenomena. Furthermore, the bubble event did not reach all the low-latitude stations. In such case, we observe only small fluctuations
within the DIX first scale on these locations. Then, we observe a peak in the DIX over these stations at 03:00 UT, within an interval of 1h40min after the peaks observed in the equatorial region. This behavior matches with the third case presented in the previous section, where the latitudinal propagation of the bubble event was of the same order of time. Therefore, we can state that the DIX also responds to the latitudinal development of plasma bubbles since the observed disturbances presented a similar





temporal pattern. Finally, we see that as the bubble event comes to an end, the DIX starts decreasing in both latitude ranges, returning to the initial non-perturbed state.

The second case (Figure 5b) shows the occurrence of plasma bubbles that were registered on the ASI and/or IONO data at São João do Cariri from 22:00 to 05:30 UT on 5-6 January 2018. Moreover, ionograms from the CAJ2M ionosonde showed spread-F signatures from 02:00 to 04:30 UT on 6
January 2015. It was not possible to see the bubble signatures on the OI 630 nm airglow images obtained over CP since there was not enough sky visibility. Furthermore, the DIX over the equatorial GNSS stations already presented an increasing behavior before 21:00 UT, which is probably associated with disturbances driven by the neutral atmosphere (as reported in Picanço et al., 2021). These ionospheric disturbances peak at around 22:30 UT and the DIX later starts to decline. Then, with the
plasma bubbles growth, the DIX over this region increases again, reaching the first peak at around 23:30 UT. At low latitudes, the DIX curves continue to show a decreasing behavior associated with perturbations prior to the plasma bubble event. It is interesting to note that this decreasing behavior ceases at around 02:20 UT, immediately after spread-F signatures are seen in the ionograms of the CAJ2M ionosonde. We notice that the DIX over the equatorial region presented a peak at around the
same time, surpassing the weakly disturbed scale (DIX under 2) (Denardini et al., 2020a). At first, we can state that the plasma bubble event was intense enough to disturb the low-latitude ionosphere for a short period since the spread-F signatures were only observed during the time of occurrence of this peak. Finally, the DIX over both latitude ranges started to decrease after the end of the plasma bubble event, remaining close to zero.

Figure 5c shows plasma bubbles registered on the ASI and/or IONO data at São João do Cariri from 22:00 to 07:00 UT on 20-21 December 2019. Additionally, ionograms from the CAJ2M ionosonde show spread-F signatures during the period from 03:40 to 06:50 UT on 21 December 2019. Since the available OI 630 nm airglow images over CP were cloudy almost all the period of study, it was not possible to observe the presence/absence of bubble signatures on it. Moreover, the DIX over the
equatorial GNSS stations shows a peak at around 22:30 UT, after the beginning of the plasma bubble growth phase. After that, small DIX fluctuations occur until the end of the plasma bubbles event. In the region of low latitudes, the DIX does not show higher values during the initial period of the bubble



development. The absence of spread-F in the ionograms obtained between 21:00 and 03:50 UT at CAJ2M indicates that the bubble did not change the TEC at low latitudes to generate strong
disturbances. However, as the irregularities start to develop spread-F in the CAJ2M ionograms (03:50 to 06:50 UT), we can see a gradual increase in DIX, which reaches a peak at 06:50 UT. This feature reinforces that the DIX is strongly related to the physical variations of the ionosphere that are measured by the ionosonde (Denardini et al., 2020b).

Finally, Figure 5d presents a plasma bubbles event registered on the ASI and/or IONO data from
22:30 to 04:40 UT at São João do Cariri on 20-21 March 2020. In addition, it was not possible to observe bubble signatures on the OI 630 nm airglow images obtained over CP since there was not enough sky visibility for that. Furthermore, the ionograms from the CAJ2M ionosonde did not show spread-F signatures during all the period of study. Nevertheless, the DIX over the equatorial latitudes started to rise at around 22:45 UT, which is 15 min after the bubble event beginning is observed on the
ASI images. On the other hand, the DIX over low latitudes did not show any abrupt variations when the equatorial ionosphere showed DIX disturbances. In the absence of spread-F on the ionograms from the CAJ2M ionosonde, it is reasonable to state that this EPBs event was not strong enough to generate TEC depletions detectable by DIX. It is noteworthy that the DIX over the equatorial GNSS stations presented a pulse-like response in the presence of plasma bubbles/spread-F, while the ionosphere over low
latitudes behaved similarly, but without the concurrent DIX response since no spread-F was registered.





**Figure 5:** Time variations of the DIX for equatorial and low-latitude stations, along with OI 630 nm airglow images taken at CA and CP, and ionograms showing the spread-F occurrence at FZA0M and CAJ2M on 23-24 February 2017 (a), 5-6 January 2018 (b), 20-21 December 2019 (c), and 20-21 March 2020 (d).





### 3.4 Overview of the ionospheric disturbances observed during solar cycle 24-25

We analyzed the DIX values calculated during each of the studied plasma bubble periods from 2013 to 2020. Specifically, we first obtained the maximum DIX values associated with the EPB-related TEC disturbances. Then, we calculated an average of these max values for each event/year, considering the four equatorial (PBCG, PBJP, PEAF, and RNNA) and low-latitude (CHPI, UBA1, MGIN, and MGVA) GNSS stations. This analysis was performed to compare the incidence of DIX peaks caused by plasma bubbles with the solar cycle 24-25 variation. In addition, Table 3 presents a summary of these maximum DIX values observed during the EPB cases presented in previous section.

**Table 3:** Summary of the maximum DIX values attributed to plasma bubbles during the period from 2013 to 2020 for all the analyzed GNSS stations: PBCG, PBJP, PEAF, RNNA, CHPI, UBA1, MGIN, and MGVA.

| Period | PBCG | PBJP | PEAF | RNNA | CHPI | UBA1 | MGIN | MGVA |
|---|---|---|---|---|---|---|---|---|
| 2013 | 0.9 | 1.2 | 1.0 | 0.9 | 1.3 | 1.3 | 1.3 | 1.4 |
| 2014 | 2.5 | 2.0 | 2.6 | 2.1 | 2.6 | 2.3 | 2.8 | 2.1 |
| 2015 | 2.2 | 3.3 | 2.4 | 2.6 | 2.0 | 1.9 | 2.0 | 1.9 |
| 2016 | 0.7 | 0.9 | 1.0 | 0.8 | 0.6 | 0.7 | 0.8 | 0.8 |
| 2017 | 1.6 | 1.5 | 1.3 | 1.6 | 0.5 | 0.4 | 0.6 | 0.4 |
| 2018 | 1.0 | 1.5 | 1.3 | 0.3 | 1.4 | 1.3 | 1.3 | 1.0 |
| 2019 | 1.0 | 1.9 | 1.4 | 0.8 | 0.2 | 0.1 | 0.1 | 0.5 |
| 2020 | 0.6 | 1.2 | 0.6 | 0.5 | 0.1 | 0.1 | 0.1 | 0.3 |

Figure 6 shows the time variation of the averaged maximum DIX values during the studied plasma bubbles events shown in Table for the equatorial (left panel) and low latitudes (right panel), along with the yearly mean total sunspot number curve (navy line).





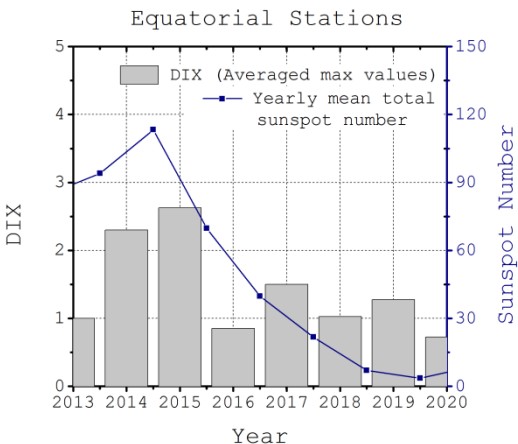 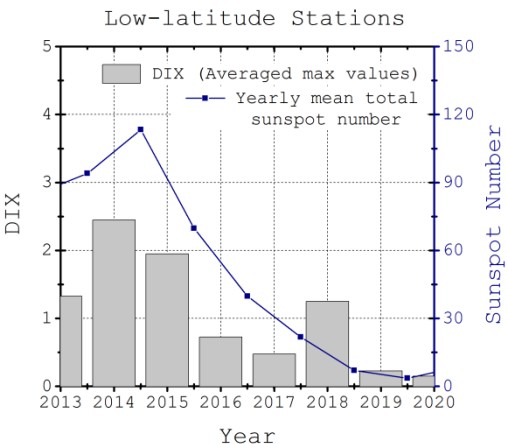

**Figure 6:** Averaged max DIX time variation during selected EPBs events observed from 2013 to 2020 over the equatorial (left panel) and low-latitude (right panel) GNSS stations along with the yearly mean total sunspot number at the same period.

From the results presented in Figure 6 and Table 3, we have noticed that the EPB-related disturbances in DIX tend to follow the trend of solar activity, which is represented by the variation of the yearly mean total sunspot number. In this context, the highest disturbances due to EPBs occurred between 2014 and 2015, over both equatorial and low-latitude GNSS stations. This period coincides with the peak of solar activity in cycle 24 (around 2014), where the mean sunspot number was approximately 113. Specifically, in the equatorial region, we observed that the intensity of the EPB-related DIX responses closely followed the variations of the solar activity level between 2013 and 2016. However, we can observe small fluctuations within the DIX scale in the period between 2017 and 2019. Besides that, the disturbances followed a decreasing average behavior until the beginning of solar cycle 25 in 2020. On the other hand, the low-latitude region showed a trend of variation much more consistent with the solar activity level, except for the plasma bubble event observed in 2018. In this event, the DIX values remained above the scale of nearby events.

Despite the observation of events with slightly variable magnitudes, TEC disturbances observed during the solar cycle 24-25 followed the trend of solar activity during most of the period studied. Therefore, the EPB-related ionospheric disturbances observed using the DIX are expected to behave according to seasonal and solar activity characteristics. However, it is important to mention that the





ionosphere degree of perturbation during these events can also be influenced by non-EPB phenomena
that may occur simultaneously with the presence of plasma bubbles.

In short, those results indicate a clear solar activity dependence of the plasma bubbles intensity
with the solar cycle. Since few studies on the solar activity variation of the plasma bubbles physical
characteristics are found in the literature, these results are important to understand how EPBs disturb the
ionosphere along the solar cycle stages. For instance, Agyei-Yeboah et al. (2019) presented a study of
the EPBs occurrence rate during the solar cycle 23-24 over the Brazilian equatorial region. The authors
stated that plasma bubbles tend to be more frequent during high solar activity, followed by moderate
and low solar activity conditions. In this context, the results presented here indicate that EPBs are also
more intense around the peak of solar cycles for both equatorial and low-latitude Brazilian regions.
Finally, this feature can be directly associated with the physical mechanisms that control the production
of electron-ion pairs in the ionosphere.

## 4 Conclusions

This paper analyzed the ionospheric total electron content responses to plasma bubbles by using
ionosonde, all-sky imager, and GNSS data over Brazilian equatorial and low-latitude stations. The
results were discussed in terms of the disturbance ionosphere index behavior under different solar
activity levels and compared to other well-established parameters found in the literature. We summarize
the conclusions below:

1. The DIX proved to detect the TEC perturbations generated by weak, moderate, and strong
plasma bubble events. In those EPB cases, the DIX presented a gradual increase as the bubbles
started to grow and tended to return to their initial non-perturbed behavior with the end of the
plasma bubble events.

2. The DIX over all studied GNSS stations also responded to the presence of spread-F on the
ionograms obtained from equatorial and low-latitudes ionosondes. In this context, the DIX
behavior matched with the intensity of spread-F, being higher or slower according to the





ionograms characteristics. Additionally, the occurrence of spread-F in all ionograms coincided with peaks observed in DIX over both equatorial and low-latitude GNSS stations.

3. We observed a delay in the DIX response time to EPBs between low-latitude and equatorial GNSS stations in some of the studied cases. For instance, this time delay was 40 minutes during the first case (2013). Therefore, we suggest that the latitudinal propagation time of the EPB-related plasma disturbances between the equatorial and low-latitude regions, which is a new feature that was observed using DIX. Therefore, the DIX can estimate the propagation time of the EPB-related ionospheric disturbances between latitude ranges.

4. The DIX also responded to the EPB eastward longitudinal propagations in some cases, where the first disturbances occurred 1 hour earlier at the western equatorial GNSS station (PEAF), which is ~200 km away from the center (PBCG) of the equatorial study area. Thus, the DIX can also estimate the propagation time of the EPB-related ionospheric disturbances between longitude ranges.

5. The contribution of neutral atmospheric effects intensified some DIX disturbances observed during the EPB periods. Since the DIX is an index that responds to several phenomena, we emphasize that the ionospheric TEC can remain disturbed even after the end of the plasma bubbles event due to other phenomena.

6. Despite observing events with slightly variable magnitudes, TEC disturbances observed during the solar cycle 24-25 followed the trend of solar activity during most of the period studied. Therefore, the EPB-related ionospheric TEC disturbances are expected to behave according to seasonal and solar activity characteristics. However, it is important to mention that the ionosphere degree of perturbation during these events can also be influenced by non-EPB phenomena that may occur simultaneously with the presence of plasma bubbles.





**Acknowledgments, Samples, and Data**

The authors thank the Embrace/INPE Space Weather Program for providing the ionosonde and All-Sky Imager data, and the Brazilian Institute of Geography and Statistics (IBGE) for providing the raw GNSS data. G. A. S. Picanço thanks CNPq/MCTIC, Brazil (Grant 132252/2017-1) and Capes/MEC, Brazil (Grant 88887.351778/2019–00). C. M. Denardini thanks CNPq/MCTIC, Brazil (Grant 303643/2017- 0). L. C. A. Resende thanks National Space Science Center (NSSC), Chinese

Academy of Sciences (CAS). C. S. Carmo thanks Capes/MEC, Brazil for supporting her Ph.D (Grant 141935/2020-0). P. F. Barbosa Neto thanks Capes/MEC, Brazil (Grant 1622967). S. S. Chen thanks CNPq/MCTIC, Brazil (Grant 134151/2017-8) and Capes/MEC, Brazil (Grant 88887.362982/2019–00). The data used in the present study are fully open and accessible in acknowledgment basis at the Embrace/INPE Program website (http://www.inpe.br/spaceweather). The sunspot data are available

online at the NOAA website (https://www.ngdc.noaa.gov/stp/solar/ssndata.html). This study was financed in part by the Coordenação de Aperfeiçoamento de Pessoal de Nível Superior - Brasil (Capes) - Finance Code 001.

**Author Contributions**

GASP conceived the study, designed the data analysis and leaded writing this manuscript.

CMD assisted to design the study and the data analysis.

PABN assisted to design the study and the data analysis.

LCAR assisted to design the study and process the ionospheric data analysis.

CSC assisted in reviewing the manuscript and discuss the results of the study.

SSC assisted to process the interplanetary data analysis and discuss the results of the study.

PFBN assisted in reviewing the manuscript and discuss the results of the study.





ERH assisted in reviewing the manuscript and discuss the results of the study.

All the authors helped to write and revise the manuscript.

## Competing interests

The authors declare that they have no conflict of interest.

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
