# Peer review of "Study of the equatorial and low-latitude TEC response to plasma bubbles during the solar cycle 24-25 over the Brazilian region using a Disturbance Ionosphere indeX"

_Annales Geophysicae, 2021_

## Referee Comment (RC1)

In this manuscript, the authors used the DIX to evaluate the ionospheric responses to EPB events from 2013 to 2020 over the Brazilian equatorial and low latitudes. Their results show that DIX is able to detect EPB-related TEC disturbances. However, the following points should be considered and improved:

1. For ROTI in this work, authors should show detailed calculation method.

2. For airglow picture in this work, in order to better compare TEC observation, the authors should map them into the geographical coordinate.

3. In the PBCG of Figure 2, the DIX shows a large value 5 at ~ 2 UT. However, the ROTI shows a small value 0.2. Similarly, the DIX of PBJB shows a large value 4 at ~ 3 UT but its ROTI shows a small value 0.1 at the same time. Same result also appears in the PEAF about 3 UT. It shows some inconsistent results between DIX and ROTI in these points. Why? The authors should explain it.

4. In Figure 3, some results are similarly to the results of Figure 2. ROTI is a small value while the DIX shows a large value at some points, such as the value of MGV at ~ 5 UT. The authors should explain it.

5. In Figure 4, it is difficult for me to distinguish between the yellow and the orange on my computer screen. I suggest that authors use contrasting colors to replace them.

6. In Figure 4 and 5, the authors showed only one picture (airglow and ionogram) in every event. As reader, based on only one airglow and ionogram, I have difficulty understand the texts of manuscript corresponding to these Figures. I suggest the authors to show more detailed airglow and ionogram pictures in each subgraph.

7. In Table 3 and Figure 6, the authors did not explain the reason why they used the maximum DIX values to compare the yearly mean total sunspot number. The yearly mean total sunspot number shows the average of solar activity in one year. However, the maximum DIX may be from only one EPB event. There is a large randomness in one EPB event. For example, the maximum DIX just appeared in one EPB event in one year and it may be caused by strong storm or others. If authors use only the maximum DIX to compare the yearly mean total sunspot number, they may get an unreal result.

8. In Figure 6, 2013 is a higher solar activity year. The mean total sunspot number of 2013 is significantly higher than 2017, 2018 and 2019. However, the DIX of 2013 is significantly lower or equal to that of 2017, 2018 and 2019 at Equatorial stations. Meanwhile, the DIX of 2013 also equals to the value of 2018 at low stations. These results disagree with the year varieties of solar activity. Why? It leads readers into confusion. The authors should explain it in detail.

---

## Author Comment (AC1)

**Responses to the Comment and/or Suggestions from Anonymous Referee #1**

**Manuscript**: angeo-2021-71
**Title**: Study of the equatorial and low-latitude TEC response to plasma bubbles during the solar cycle 24-25 over the Brazilian region using a Disturbance Ionosphere indeX
**Authors**: Picanço et al.
**Reviewer**: Anonymous Referee #1

Comments on the manuscript entitled "Study of the equatorial and low-latitude TEC response to plasma bubbles during the solar cycle 24-25 over the Brazilian region using a Disturbance Ionosphere indeX" by Picanço et al. submitted to the Annales Geophysicae journal.

In this manuscript, the authors used the DIX to evaluate the ionospheric responses to EPB events from 2013 to 2020 over the Brazilian equatorial and low latitudes. Their results show that DIX is able to detect EPB-related TEC disturbances. However, the following points should be considered and improved:

*Our response:*
*First, we would like to take this opportunity to thank Reviewer #1 for his/her time spent evaluating our contribution. In the following lines, we provide the specific answers to each specific point risen by the reviewer*

1. For ROTI in this work, authors should show detailed calculation method.

*Our response:*
*We agree with the reviewer's suggestion. In this regard, we will now add a full explanation of the methodology used to calculate ROTI in the revised version of the paper.*

2. For airglow picture in this work, in order to better compare TEC observation, the authors should map them into the geographical coordinate.

*Our response:*
*We thank Reviewer #1 for his/her suggestion. Indeed, such a modification will facilitate comparing airglow and TEC data. Therefore, such changes will be made in the figures indicated.*

3. In the PBCG of Figure 2, the DIX shows a large value 5 at ~ 2 UT. However, the ROTI shows a small value 0.2. Similarly, the DIX of PBJB shows a large value 4 at ~ 3 UT but its ROTI shows a small value 0.1 at the same time. Same result also appears in the PEAF about 3 UT. It shows some inconsistent results between DIX and ROTI in these points. Why? The authors should explain it.

*Our response:*
*We thank the reviewer for raising this important observation. We noticed a mistake in the time format of the DIX values presented in figures 2 and 3, which will be corrected in the revised figures. This caused the EPB-related DIX peaks to appear displaced in relation to the ROTI peaks in both figures. Therefore, we attach as an example a figure with the DIX and ROTI corrected curves for PBCG, along with the vTEC and sTEC curves from which we have derived DIX and ROTI, respectively. Then, we can observe that the first DIX peak is coherent with the time of occurrence of the first ROTI peak. Additionally, we would like to emphasize that the DIX is not an index specifically made to detect small-scale irregularities, such as the ROTI. The DIX is an index that responds to TEC variations in general, whether caused by internal (eg, EPBs) or external (eg, magnetic storms) sources. In this regard, the DIX peaks*

*after 04:30 UT (figure below) occur probably due to TEC disturbances caused by other ionospheric effects not associated with plasma bubbles.*

[Figure]

4. In Figure 3, some results are similarly to the results of Figure 2. ROTI is a small value while the DIX shows a large value at some points, such as the value of MGV at ~ 5 UT. The authors should explain it.

*Our response:*

*As it was made in the previous topic, we attach here an example of corrected DIX and ROTI curves for MGV, along with the vTEC and sTEC curves from which we have derived DIX and ROTI, respectively. In this figure, we observe that both EPB-related DIX and ROTI peaks occur around the same time interval (03:30 UT - 03:45 UT). In addition, two peaks appear later only in the DIX, being those possibly associated with the occurrence of TEC variations that are not related to plasma bubbles.*

*We are grateful for the reviewer's comment, which may avoid misunderstanding in the revised version of the present manuscript.*

[Figure]

30-01-2014 (MGV, PRN 23)

5. In Figure 4, it is difficult for me to distinguish between the yellow and the orange on my computer screen. I suggest that authors use contrasting colors to replace them.
*Our response:*
*We thank Reviewer #1 for the relevant observation. We will select contrasting colors so that the results are better visualized.*

6. In Figure 4 and 5, the authors showed only one picture (airglow and ionogram) in every event. As reader, based on only one airglow and ionogram, I have difficulty understand the texts of manuscript corresponding to these Figures. I suggest the authors to show more detailed airglow and ionogram pictures in each subgraph.
*Our response:*
*We thank the reviewer for the suggestion. We will include more airglow and ionogram images in the results.*

7. In Table 3 and Figure 6, the authors did not explain the reason why they used the maximum DIX values to compare the yearly mean total sunspot number. The yearly mean total sunspot number shows the average of solar activity in one year. However, the maximum DIX may be from only one EPB event. There is a large randomness in one EPB event. For example, the maximum DIX just appeared in one EPB event in one year and it may be caused by strong storm or others. If authors use only the maximum DIX to compare the yearly mean total sunspot number, they may get an unreal result.
*Our response:*
*We understand the reviewer's concern in raising this doubt. Perhaps it was unclear, but all periods studied are geomagnetically quiet (kp<=3). Unfortunately, we did not specify this in the text. Therefore, we will include a more detailed description of the methodology for selecting bubble events so that this doubt can be clarified. In short, we intended to compare the intensity of the plasma bubbles (DIX max) with the variation of solar activity (sunspot number). We will include a better explanation in the revised version of the manuscript. Thanks.*

8. In Figure 6, 2013 is a higher solar activity year. The mean total sunspot number of 2013 is significantly higher than 2017, 2018 and 2019. However, the DIX of 2013 is significantly lower or equal to that of 2017, 2018 and 2019 at Equatorial stations. Meanwhile, the DIX of

2013 also equals to the value of 2018 at low stations. These results disagree with the year varieties of solar activity. Why? It leads readers into confusion. The authors should explain it in detail.

*Our response:*

*We thank the reviewer for raising this point. Indeed, the magnitude of EPB-related DIX disturbances tends to follow the temporal trend of solar activity in most cases. However, as we have analyzed data from one event per year, the specific disturbances observed during de 2013 EPB event tended to keep the maximum DIX under the scale 2. Specifically, the 2013 plasma bubble event was less intense than the others, so DIX showed smaller-scale disturbances. In this regard, such a weak EPB event caused the 2013 maximum DIX to be smaller than DIX values observed in 2017, 2018, and 2019. This feature can be seen in Figure 4, where the DIX is around the scale of 1 at all GNSS stations. Thus, we will include a better explanation for this question in the revised version of the paper.*

*Finally, we would like to take this opportunity to thank the reviewer for kindly evaluating our paper helping to greatly improve its quality.*

---

## Author Comment (AC3)

**Responses to the Comment and/or Suggestions from Anonymous Referee #2**

**Manuscript**: angeo-2021-71
**Title**: Study of the equatorial and low-latitude TEC response to plasma bubbles during the solar cycle 24-25 over the Brazilian region using a Disturbance Ionosphere indeX
**Authors**: Picanço et al.
**Reviewer**: Anonymous Referee #2

Comments on the manuscript entitled "Study of the equatorial and low-latitude TEC response to plasma bubbles during the solar cycle 24-25 over the Brazilian region using a Disturbance Ionosphere indeX" by Picanço et al. submitted to the Annales Geophysicae journal.

The authors use Disturbance Ionosphere index (DIX) to evaluate the ionospheric responses to Equatorial Plasma Bubbles (EPBs) between 2013 and 2020 over the Brazilian sector. The results of the DIX were compared concurrent EPB observations from ionosonde and All-Sky Imager data. The authors show that the DIX was able to detect EPB-related disturbances both in terms of intensity and occurrence times. Finally, it was shown that the magnitude of the disturbances depends on solar activity. The major points of this study are quite interesting and useful. However, there are some comments that the authors need to consider to strengthen the points presented in this work. Please find the comments below.

*Our response:*

*We would like to thank Reviewer #2 for his/her time spent evaluating our paper. In the following lines, we provide the specific answers to each point raised by the reviewer. We included the information in the revised version of the paper as suggested by the reviewer.*

Line 89: There is the need to state the basis for selecting the EPBs under this study in the section 2, or are these the only EPB events between 2013 and 2020.

*Our response:*

*We thank the reviewer for highlighting this point. We will include all information in the revised version of the paper as suggested by the reviewer. We selected EPB events that occurred near the summer of each year, aiming for a better comparison among them. In this regard, we also based our selection on data availability, taking into account the simultaneity of GNSS, Ionosonde, and All-Sky Imager observations. We included these details in the revised version of the paper as suggested by the reviewer.*

Line 97: There is no $TEC_k$ in equation (1), the authors should revise this equation.

*Our response:*

*We thank Reviewer #2 for the relevant observation. We corrected that in the revised version of the paper.*

*The new equation is:*

$$DIX_k(t) = \left| \frac{\alpha\left(\Delta TEC_k(t)/TEC_k^{Qd}(t)\right) + \Delta TEC_k(t)}{\beta} \right|, \qquad (1)$$

Line 115: It may be useful to have a third column on Table 1 describing the implication of column 2.

*Our response:*

*We thank Reviewer #2 for such a pertinent suggestion. However, we comprehend that this knowledge goes beyond the scope of the present paper. On the other hand, such a suggestion is relevant for evaluating DIX applications within a space weather context. Therefore, it surely will be put into practice in our future manuscript since it is possible to compare GNSS positioning errors with the DIX scale. In the present work, we have normalized the ionospheric disturbances into a scale ranging from 0 to 5, considering the level 5 as the highest disturbed state observed in all analyzed cases.*

Line 252-254: ''Such effect can be explained by the presence of the Equatorial Ionization Anomaly (EIA) southern crest, which produces a large amount of plasma in this region, making the TEC percentage changes also higher'' …. Did the authors confirm this inference from the TEC observations since some other cases did not follow this inference?

*Our response:*

*We thank Reviewer #2 for raising this point. The equatorial ionization anomaly (EIA) is mainly characterized by a plasma trough in the equatorial region, which is flanked by two crests in low latitudes, generally occurring at around ±15° dip latitude. The EIA crests present a high day-to-day variability, which can suddenly change its position (Takahashi et al., 2016). Since we are using data from four nearby GNSS stations at low latitudes, the EIA south crest might be above or below those stations over time. In some cases, DIX values at low latitudes might be higher than those over the magnetic equator, considering that these stations are under influence of the southern EIA crest and can be affected by its dynamics. We included this explanation in the text to clarify to the reader.*

*Takahashi, H., Wrasse, C. M., Denardini, C. M., Pádua, M. B., de Paula, E. R., Costa, S. M. A., et al. (2016). Ionospheric TEC weather map over south America. Space Weather, 14, 937–949. https://doi.org/10.1002/2016SW001474.*

Line 237-239: The reason for the selection of the specific ASI snapshots and the ionograms under Figures 4 & 5 should be mentioned.

*Our response:*

*We thank the reviewer for pointing this out. We selected snapshots that could facilitate the visibility of the plasma bubble signatures. In this regard, ASI images can be affected by other atmospheric phenomena (eg clouds, rain), and the spread-F observed in the ionograms tends to be more intense at the peak of the bubble passage over the ionosonde field of view. We considered those factors when selecting the images. We included more details about the referred data selection in the revised version of the paper.*

Line 237: There is the need for the authors to state the possible reason(s) why at times the response of the DIX is simultaneously observed at both the equatorial and low latitude stations while it's not at other times, does the authors have any explanation for this?

*Our response:*

*We thank Reviewer #2 for raising such a relevant point. We agree with the observation that in some EPB cases the DIX responses are not observed simultaneously at equatorial and low-latitude GNSS stations.*

*In this context, we start arguing from the concepts presented in the previous question, which deals with the intensity of plasma depletions in different latitude ranges. If we take the case presented in Figure 4.d as an example, we notice that the bubble signature did not fully reach the field of view of the Cachoeira Paulista All-Sky Imager. As a consequence of this scenario, DIX values at low-latitude stations remained lower since only partial plasma bubble signatures have reached this region in comparison to the equatorial ones. Moreover, the opposite situation does not occur, as we did not detect cases in which DIX peaks associated with EPBs were detected only over low-latitude GNSS stations.*

*Therefore, those cases where DIX peaks are not observed simultaneously over equatorial and low-latitudes stations are possibly due to EPB events that have not fully developed to the low-latitude ionospheric region.*

*All this information was included in the new version of the manuscript.*

Line 355: Change '6 January 2015' to 6 January 2018.

*Our response:*

*We apologize for that. We have corrected it.*

Line 394: It is not quite clear what the authors meant by '… while the ionosphere over low latitudes behaved similarly ...' This is with respect to what?

*Our response:*

*We apologize for that little mistake. We have corrected it as follows.*

*"It is noteworthy that the DIX over the equatorial GNSS stations presented a pulse-like response in the presence of plasma bubbles/spread-F, while the ionosphere over low latitudes behaved similarly to the expected for non-disturbed periods, without the concurrent DIX response since no spread-F was registered."*

Line 415: For the presentation in Figure 6, it will be better to include more EPB events between 2013 and 2020 if there are, for more reliable statistics. The authors can also show the error bars of the standard deviation for each year.

*Our response:*

*We thank reviewer #2 for his/her suggestion. We included more information in the graph as suggested.*

Line 446: change 'those results' to 'these results'.

*Our response:*
*We have corrected it. Thank you.*

Line 454-455: Can the authors expound on how they arrived at this conclusion: '*Finally, this feature can be directly associated with the physical mechanisms that control the production of electron-ion pairs in the ionosphere*'.

*Our response:*
*We thank the reviewer for raising this point. We have removed such unnecessary sentences.*

Line 474: The authors can mention the specific event being referred to here.

*Our response:*
*We have included the information as the reviewer suggested (see below). Thank you.*

*"We observed a delay in the DIX response time to EPBs between low-latitude and equatorial GNSS stations in some of the studied cases (2013, 2014, and 2016). For instance, this time delay was 40 minutes during the first case (2013). Therefore, we suggest it as the latitudinal propagation time of the EPB-related plasma disturbances between equatorial and low-latitude regions, which is a new feature observed using DIX. Therefore, the DIX can estimate the propagation time of the EPB-related ionospheric disturbances between latitude ranges."*

Line 484: Change '*The contribution of neutral atmospheric effects intensified some DIX disturbances observed during the EPB periods*' to '*The contribution of neutral atmospheric effects may have intensified some DIX disturbances observed during the EPB periods*'.

*Our response:*
*We have modified that. Thank you.*

The authors did not provide full information of some of references, consider adding the issue and volume numbers to them: line 534, line 541, line 558, line 562, line 566, line 580.

*Our response:*
*This part of the manuscript is now properly updated.*

*Finally, we would like to take this opportunity to thank the reviewer for kindly evaluating our paper and helping to improve its quality.*

---

## Author Response (AR1)

**Responses to the Comment and/or Suggestions from Anonymous Referee #1**

**Manuscript**: angeo-2021-71
**Title**: Study of the equatorial and low-latitude TEC response to plasma bubbles during the solar cycle 24-25 over the Brazilian region using a Disturbance Ionosphere index
**Authors**: Picanço et al.
**Reviewer**: Anonymous Referee #1

Comments on the manuscript entitled "Study of the equatorial and low-latitude TEC response to plasma bubbles during the solar cycle 24-25 over the Brazilian region using a Disturbance Ionosphere indeX" by Picanço et al. submitted to the Annales Geophysicae journal.

In this manuscript, the authors used the DIX to evaluate the ionospheric responses to EPB events from 2013 to 2020 over the Brazilian equatorial and low latitudes. Their results show that DIX is able to detect EPB-related TEC disturbances. However, the following points should be considered and improved:

*Our response:*

*First, we would like to take this opportunity to thank Reviewer #1 for his/her time spent evaluating our contribution. In the following lines, we provide the specific answers to each specific point risen by the reviewer*

1. For ROTI in this work, authors should show detailed calculation method.

*Our response:*

*We agree with the reviewer's suggestion. In this regard, we added a full explanation of the methodology used to calculate ROTI in the revised version of the paper (see Section 2.2).*

2. For airglow picture in this work, in order to better compare TEC observation, the authors should map them into the geographical coordinate.

*Our response:*

*We thank Reviewer #1 for his/her suggestion. We performed these modifications in all the figures containing ASI images (see Figures 1-4 in the revised paper and Figures S3, S5, S7, S9, S11, S13, S15, and S17 in the supplementary material).*

3. In the PBCG of Figure 2, the DIX shows a large value 5 at ~ 2 UT. However, the ROTI shows a small value 0.2. Similarly, the DIX of PBJB shows a large value 4 at ~ 3 UT but its ROTI shows a small value 0.1 at the same time. Same result also appears in the PEAF about 3 UT. It shows some inconsistent results between DIX and ROTI in these points. Why? The authors should explain it.

*Our response:*

*We thank the reviewer for raising this point. We noticed a mistake in the time format of the DIX values presented in Figures 2 and 3, which we have fixed in the revised paper. This mistake caused the EPB-related DIX peaks to appear displaced in relation to the ROTI peaks in both figures. As a matter of example, we attach here a figure with the DIX and ROTI corrected curves for PBCG, along with the VTEC and STEC curves from which we have derived DIX and ROTI, respectively (also included in the supplementary material, Figure S1).*

*Then, we can observe that the first DIX peak is coherent with the time of occurrence of the first ROTI peak. In addition, we would like to emphasize that the DIX is not an index specifically to detect small-scale irregularities, such as the ROTI. The DIX is an index that responds to general TEC variations caused by internal (e.g., EPBs) and/or external (e.g., magnetic storms) sources. We added this information in lines 243-249.*

*Therefore, it is reasonable to state that the DIX peaks after 04:30 UT (figure below) may have occurred probably due to the sum of all current TEC disturbances, including those caused by other ionospheric effects not necessarily associated with plasma bubbles.*

[Figure]

30-01-2014 (MGV, PRN 23)

4. Figure 3, some results are similarly to the results of Figure 2. ROTI is a small value while the DIX shows a large value at some points, such as the value of MGV at ~ 5 UT. The authors should explain it.

*Our response:*

*As we did in the previous topic, we attach here an example of corrected DIX and ROTI curves for MGV, along with the VTEC and STEC curves from which we have derived DIX and ROTI, respectively (included in the supplementary material, Figure S2). In that figure, we observe that the EPB-related DIX and ROTI peaks occur around the same time interval (03:30 UT–03:45 UT). In addition, two peaks appear later only in the DIX, being those possibly associated with the occurrence of TEC variations that may not be necessarily related to plasma bubbles. Additional research is necessary to explain the origin of these non-EPB peaks. However, this need is not within the scope of this article. We included this information in lines 243-249.*

*We are grateful for the reviewer's comment, which may avoid misunderstanding in the revised version of the present manuscript.*

[Figure]

30-01-2014 (PBCG, PRN 23)

5.  In Figure 4, it is difficult for me to distinguish between the yellow and the orange on mycomputer screen. I suggest that authors use contrasting colors to replace them.
*Our response:*
*We selected contrasting colors so that the results are better visualized (see new Figures 4 and 5).*

6.  In Figure 4 and 5, the authors showed only one picture (airglow and ionogram) in every event. As reader, based on only one airglow and ionogram, I have difficulty understand the texts of manuscript corresponding to these Figures. I suggest the authors to show more detailed airglow and ionogram pictures in each subgraph.
*Our response:*
*We thank the reviewer for such a pertinent suggestion. We added an extended collection of ASI and IONO images in the supplementary material (Figures S3 to S18).*

7.  In Table 3 and Figure 6, the authors did not explain the reason why they used the maximum DIX values to compare the yearly mean total sunspot number. The yearly mean total sunspot number shows the average of solar activity in one year. However, the maximum DIX may be from only one EPB event. There is a large randomness in one EPB event. For example, the maximum DIX just appeared in one EPB event in one year and it may be causedby strong storm or others. If authors use only the maximum DIX to compare the yearly mean total sunspot number, they may get an unreal result.
*Our response:*
*We understand the reviewer's concern in raising this doubt. Perhaps it was unclear, but all periods studied are geomagnetically quiet (Kp<=3). Specifically, we ensured using GNSS, ASI, and IONO data obtained only during EPB-events within no or insignificant magnetic activity. Furthermore, we selected the maximum DIX value because it represents the bubble intensity within the scale of the ionospheric states (Table 1).Thus, we aimed to compare the bubble intensity (max DIX) with the solar-cycle temporal behavior (sunspot number). Lastly, we included a more detailed description of the methodology for selecting bubble events so that this doubt can be clarified (lines179-183).*

8. In Figure 6, 2013 is a higher solar activity year. The mean total sunspot number of 2013 is significantly higher than 2017, 2018 and 2019. However, the DIX of 2013 is significantly lower or equal to that of 2017, 2018 and 2019 at Equatorial stations. Meanwhile, the DIX of 2013 also equals to the value of 2018 at low stations. These results disagree with the year varieties of solar activity. Why? It leads readers into confusion. The authors should explain it in detail.

*Our response:*

*We thank the reviewer for raising this point. Indeed, the magnitude of EPB-related DIX disturbances tends to follow the temporal trend of solar activity in most cases. However, as we have analyzed data from one event per year, the specific disturbances observed during de 2013 EPB event tended to keep the maximum DIX under the scale 2. Specifically, the 2013 plasma bubble event was weaker than the others, so DIX showed smaller-scale disturbances. Such a weak EPB event caused the 2013 maximum DIX to be smaller than the DIX values observed in 2017, 2018, and 2019. This feature can be seen in Figure 4, where the DIX is around the scale of 1 at all GNSS stations. Thus, we redid this analysis using more events per year to mitigate the effects of outliers. We included a better explanation for this question in the revised version of the paper (please see Section 3.4).*

*Finally, we would like to take this opportunity to thank the reviewer for kindly evaluating ourpaper helping to greatly improve its quality.*

**Manuscript**: angeo-2021-71
**Title**: Study of the equatorial and low-latitude TEC response to plasma bubbles during the solar cycle 24-25 over the Brazilian region using a Disturbance Ionosphere index
**Authors**: Picanço et al.
**Reviewer**: Anonymous Referee #2

Comments on the manuscript entitled "Study of the equatorial and low-latitude TEC response to plasma bubbles during the solar cycle 24-25 over the Brazilian region using a Disturbance Ionosphere indeX" by Picanço et al. submitted to the Annales Geophysicae journal.

The authors use Disturbance Ionosphere index (DIX) to evaluate the ionospheric responses to Equatorial Plasma Bubbles (EPBs) between 2013 and 2020 over the Brazilian sector. The results of the DIX were compared concurrent EPB observations from ionosonde and All-Sky Imager data. The authors show that the DIX was able to detect EPB-related disturbances both in terms of intensity and occurrence times. Finally, it was shown that the magnitude of the disturbances depends on solar activity. The major points of this study are quite interesting and useful. However, there are some comments that the authors need to consider to strengthen the points presented in this work. Please find the comments below.

*Our response:*

*We would like to thank Reviewer #2 for his/her time spent evaluating our paper. In the following lines, we provide the specific answers to each point raised by the reviewer. We included the information in the revised version of the paper as suggested by the reviewer.*

Line 89: There is the need to state the basis for selecting the EPBs under this study in the section 2, or are these the only EPB events between 2013 and 2020.

*Our response:*

*We thank the reviewer for highlighting this point. We selected EPB events that occurred near the summer of each year, aiming for a better comparison among them. In this regard, we also based our selection on data availability, taking into account the simultaneity of GNSS, Ionosonde, and All-Sky Imager observations. Finally, we used only data from periods with no significant magnetic activity (Kp<=3) to avoid the effects of magnetic disturbances on DIX. We included these details in the revised version of the paper as suggested by the reviewer (lines 179-183).*

Line 97: There is no TEC$_k$ in equation (1), the authors should revise this equation.

*Our response:*

*We thank Reviewer #2 for the relevant observation. We corrected that in the revised version of the paper.*

*The correct equation is:*

$$\text{DIX}_k(t) = \left| \frac{\alpha \left( \Delta TEC_k(t) / TEC_k^{Qd}(t) \right) + \Delta TEC_k(t)}{\beta} \right|, \qquad (1)$$

Line 115: It may be useful to have a third column on Table 1 describing the implication of column 2.

*Our response:*

*We thank Reviewer #2 for such a pertinent suggestion. However, we comprehend that this knowledge goes beyond the scope of the present paper once it requires more extensive studies. On the other hand, such a suggestion is relevant for evaluating DIX applications within a space weather context. Therefore, it will surely be put into practice in our future manuscript since it is possible to compare GNSS positioning errors with the DIX scale. In the present work, we have normalized the ionospheric disturbances into a scale ranging from 0 to 5, considering level 5 as the most disturbed state observed in all analyzed cases.*

Line 252-254: ''Such effect can be explained by the presence of the Equatorial Ionization Anomaly (EIA) southern crest, which produces a large amount of plasma in this region, making the TEC percentage changes also higher'' …. Did the authors confirm this inference from the TEC observations since some other cases did not follow this inference?

*Our response:*

*We thank Reviewer #2 for raising this point. The equatorial ionization anomaly (EIA) is mainly characterized by a plasma trough in the equatorial region, which is flanked by two crests in low latitudes, generally occurring at around ±15° dip latitude. The EIA crests present a high day-to-day variability, which can suddenly change its position (Takahashi et al., 2016). Since we are using data from four nearby GNSS stations at low latitudes, the EIA south crest might be above or below those stations over time. In some cases, DIX values at low latitudes might be higher than those over the magnetic equator, considering that these stations are under influence of the southern EIA crest and can be affected by its dynamics. We included this explanation in the text to clarify to the reader (lines 293-300).*

*Takahashi, H., Wrasse, C. M., Denardini, C. M., Pádua, M. B., de Paula, E. R., Costa, S. M. A., et al. (2016). Ionospheric TEC weather map over south America. Space Weather, 14, 937–949. https://doi.org/10.1002/2016SW001474.*

Line 237-239: The reason for the selection of the specific ASI snapshots and the ionograms under Figures 4 & 5 should be mentioned.

*Our response:*

*We thank the reviewer for pointing this out. We selected snapshots that could facilitate the visibility of the plasma bubble signatures. In this regard, ASI images can be affected by other atmospheric phenomena (e.g., clouds, rain), and the Spread-F observed in the ionograms tends to be more intense at the peak of the bubble passage over the ionosonde field of view. We considered those factors when selecting the images. We included more details about the referred data selection in the revised version of the paper (lines 271-275).*

Line 237: There is the need for the authors to state the possible reason(s) why at times the response of the DIX is simultaneously observed at both the equatorial and low latitude stations while it's not at other times, does the authors have any explanation for this?
*Our response:*
*We thank Reviewer #2 for raising such a relevant point. We agree with the observation that in some EPB cases, the time response of the DIX is not observed simultaneously at equatorial and low-latitude GNSS stations.*

*We attribute this feature to the displacement velocity of the plasma depletion in different latitude ranges. In this regard, Dabas et al. (1992) calculate the EPB latitudinal velocity from the time difference of depletions observed between an equatorial and a low-latitude station. Bringing this concept to the context presented here, we take the case shown in Figure 4.d as an example. We observed 20- to 50-min time delays between the EPB-peaks in DIX over both latitude ranges. In addition, the bubble signature did not fully reach the CP-ASI field of view. Therefore, we suggest that this bubble event developed slower than the others, where the time delays were not significant. Consequently, DIX values at low-latitude stations remained lower since only partial plasma bubble signatures reached this region, and this event developed slower.*

*All this information was included in the new version of the manuscript (lines 378-387).*

*Dabas, R. S., Banerjee, P. K., Bhattacharya, S., Reddy, B. M., and Singh, J. (1992). Study of equatorial plasma bubble dynamics using GHz scintillation observations in the Indian sector. Journal of Atmospheric and Terrestrial Physics, 54(7-8), 893–901. doi:10.1016/0021-9169(92)90056-q.*

Line 355: Change '6 January 2015' to 6 January 2018.
*Our response:*
*We have corrected it (line 424).*

Line 394: It is not quite clear what the authors meant by '… while the ionosphere over low latitudes behaved similarly ...' This is with respect to what?
*Our response:*
*We apologize for that little mistake. We have corrected it as follows.*

*"It is noteworthy that the DIX over the equatorial GNSS stations presented a pulse-like response in the presence of plasma bubbles/spread-F, while the ionosphere over low latitudes behaved similarly to the expected for non-disturbed periods, without the concurrent DIX response since no spread-F was registered." (lines 462-465).*

Line 415: For the presentation in Figure 6, it will be better to include more EPB events between 2013 and 2020 if there are, for more reliable statistics. The authors can also show the error bars of the standard deviation for each year.
*Our response:*
*We thank reviewer #2 for his/her suggestion. We included more information as suggested (see Table 3 and Figure 6).*

Line 446: change 'those results' to 'these results'.
*Our response:*
*We have corrected it (line 546). Thank you.*

Line 454-455: Can the authors expound on how they arrived at this conclusion: 'Finally, this feature can be directly associated with the physical mechanisms that control the production of electron-ion pairs in the ionosphere'.
*Our response:*
*We thank the reviewer for raising this point. We have removed such unnecessary sentences.*

Line 474: The authors can mention the specific event being referred to here.
*Our response:*
*We have included the information as the reviewer suggested (see below). Thank you.*

*"We observed a delay in the DIX response time to EPBs between low-latitude and equatorial GNSS stations in some of the studied cases (2013, 2014, and 2016). For instance, this time delay was 40 minutes during the first case (2013). Therefore, we suggest it as the latitudinal propagation time of the EPB-related plasma disturbances between equatorial and low-latitude regions, which is a new feature observed using DIX. Therefore, the DIX can estimate the propagation time of the EPB-related ionospheric disturbances between latitude ranges." (lines 571-576).*

Line 484: Change 'The contribution of neutral atmospheric effects intensified some DIX disturbances observed during the EPB periods' to 'The contribution of neutral atmospheric effects may have intensified some DIX disturbances observed during the EPB periods'.
*Our response:*
*We have modified that (line 582). Thank you.*

The authors did not provide full information of some of references, consider adding the issue and volume numbers to them: line 534, line 541, line 558, line 562, line 566, line 580.
*Our response:*
*This part of the manuscript is now properly updated (lines 632, 639, 659, 663, 668, and 681).*

*Finally, we would like to take this opportunity to thank the reviewer for kindly evaluating our paper and helping to improve its quality.*